# Dedifferentiated Endometrial Carcinoma: A Rare Aggressive Neoplasm-Clinical, Morphological and Immunohistochemical Features

**DOI:** 10.3390/cancers15215155

**Published:** 2023-10-26

**Authors:** Giovanna Giordano, Elena Ferioli, Debora Guareschi, Alessandro Tafuni

**Affiliations:** Department of Medicine and Surgery, Pathology Unit, University of Parma, Viale A. Gramsci, 14, 43126 Parma, Italy; elena.ferioli@unipr.it (E.F.); alessandro.tafuni@unipr.it (A.T.)

**Keywords:** endometrial dedifferentiated/undifferentiated carcinoma, the Cancer Genome Atlas (TCGA) Research Network, mismatch repair (MMR) gene mutations, DNA Polymerase Epsilon (POLE), Switch/Sucrose Non-Fermentable (SWI/SNF) complex, Tp53 mutations, immune checkpoint inhibitors

## Abstract

**Simple Summary:**

In a neoplasm, dedifferentiation is characterised by the presence of a high-grade neoplasm which can occur de novo, be juxtaposed to, or arise as a recurrence of a previously well-differentiated tumour. Usually, this occurrence results in mesenchymal neoplasms. In epithelial malignant neoplasms, dedifferentiation has been observed in salivary gland carcinomas including adenoid cystic carcinoma, mucoepidermoid carcinoma, myoepithelial carcinoma, and acinic cell carcinoma. In addition, dedifferentiated carcinomas have been reported in the pancreas and in the gastrointestinal and urinary tracts. In the female genital tract, dedifferentiated carcinoma have been described in the endometrium and ovary. Histologically, this entity is characterised by both low-grade endometrioid carcinoma and a solid undifferentiated component. It is especially important to recognise this subtype of the malignancy due to its fulminant clinical outcomes and a poorer prognosis than high-grade endometrioid carcinoma. From a review of the literature, we have extracted clinical, morphological, and immunohistochemical data useful for an accurate diagnosis and prognosis of this rare endometrial malignancy.

**Abstract:**

Dedifferentiated endometrioid adenocarcinoma is characterised by the coexistence of an undifferentiated carcinoma and a low-grade endometrioid adenocarcinoma. The low-grade component in this subtype of endometrial carcinoma is Grade 1 or 2 according to the Federation of Gynaecology and Obstetrics (FIGO) grading system. The coexistence of low-grade endometrial carcinoma and solid undifferentiated carcinoma can cause diagnostic problems on histological examination. In fact, this combination can often be mistaken for a more common Grade 2 or Grade 3 endometrial carcinoma. Therefore, this subtype of uterine carcinoma can often go under-recognised. An accurate diagnosis of dedifferentiated endometrial carcinoma is mandatory because of its poorer prognosis compared to Grade 3 endometrial carcinoma, with a solid undifferentiated component that can amount to as much as 20% of the entire tumour. The aim of this review is to provide clinical, immunohistochemical, and molecular data to aid with making an accurate histological diagnosis and to establish whether there are any findings which could have an impact on the prognosis or therapeutic implications of this rare and aggressive uterine neoplasm.

## 1. Introduction

Dedifferentiation represents the presence of a high-grade neoplasm which can occur de novo or be juxtaposed to, or arise as a recurrence of a previously well-differentiated neoplasm. Dedifferentiation in a malignant neoplasm can be considered as a histological indicator of tumour progression. Moreover, this process proves that some malignant neoplasms have a plasticity, in which tumour cells lose their specialized properties and take less differentiated phenotypes reminiscent of early embryonic development or regenerative processes [1]. Dedifferentiation is often associated with increased tumour cell invasiveness and drug resistance [2,3,4] and has been recognized in many malignant epithelial neoplasms such as salivary gland carcinoma, adenoid cystic carcinoma, mucoepidermoid carcinoma, and myoepithelial carcinoma [5]. In addition, dedifferentiated carcinomas have been observed in the gastro-intestinal tract, pancreas, and urinary tract [6,7,8].

Endometrial dedifferentiated carcinoma (EDC), initially named ‘dedifferentiating endometrial carcinoma’, was reported in 1989 by Tenti et al. as a subtype of a well-differentiated carcinoma which had evolved into a higher-grade carcinoma after chemotherapy [9]. Later, in 2006, Silva et al. coined the term ‘endometrial dedifferentiated carcinoma’ (EDC) to describe an aggressive carcinoma characterised by the simultaneous presence of an undifferentiated carcinoma and endometrial carcinoma of a low grade (i.e., Grade 1 or Grade 2) [10]. In 2003, the World Health Organization (WHO) classification defined undifferentiated carcinoma as a malignancy lacking any evidence of differentiation [11]. In the 2014 WHO definition, the undifferentiated component of Endometrial EDC was described as a monomorphic neoplasm that resembled a lymphoma, plasmacytoma, high-grade endometrial stromal sarcoma, or small cell carcinoma [12,13]. EDC was considered as a rare subtype of an endometrial carcinoma, although the incidence rate was not well established. However, the incidence of endometrial undifferentiated carcinoma (EUC) is known to range from 1 to 9% [13,14]. Moreover, several retrospective studies have demonstrated that 37 to 87% of EUC were admixed with low-grade endometrial adenocarcinoma [10,13,15]. Due to the concomitant presence of a low-grade endometrial component and an undifferentiated component, EDC can often be misdiagnosed as a Federation of Gynaecology and Obstetrics (FIGO) Grade 2 or Grade 3 endometrial carcinoma [16,17]. The distinction of these different entities is of paramount importance because EDC, as was already emphasized by Silva et al., has a poorer prognosis, even when the UC component represents only 20% of the entire neoplasm [10]. In fact, this malignancy is usually diagnosed at an advanced development stage and presents resistance to conventional chemotherapy [10].

The aim of this review was to report pathological data of this rare and lethal malignancy focusing on its diagnostic and prognostic features. In addition, we also evaluated the latest research regarding the genetic alterations that this neoplasm presents in order to identify potential molecular targeting therapies. 

### 1.1. Clinical Features 

Regarding age in different studies, this neoplasm occurs during the sixth and seventh decades [18,19]. However, in some series, the patients’ age ranged from 30–84 years and many patients were <40 years [20]. Clinically, the neoplasms were characterised via vaginal bleeding. The neoplasm often presented an advanced FIGO development stage [15,18,19,21,22,23]. The clinical course in many instances was fulminant [24] and sometimes at diagnosis the neoplasm was characterised via simultaneous cerebellar and adrenal metastases [25]. In some cases, the metastases were observed in bones [26] while recurrences were also observed in bones and other organs, such as the supraclavicular lymph nodes, vagina, omentum, and peritoneum, despite neoadjuvant chemotherapy [27]. In the series reported by Goh et al. it seemed that long disease-free survival was related to a small proportion of undifferentiated component in the primary neoplasm [27]. As emphasized by Yokomizo et al., in some cases of EDC, it is possible to observe a family history of colon cancer in a first-degree relative and a loss of DNA mismatch-repair protein (MMR) expression [28]. Thus, it is also important in this subtype of endometrial carcinoma to evaluate a counselling procedure to identify the risk of Lynch syndrome. This syndrome is an autosomal dominant inherited cancer susceptibility, which is associated with germline mutations in one set of MMR genes (*MLH1*, *MSH2*, *MSH6*, and *PMS*) [29].

### 1.2. The Main Morphological and Immunohistochemical Features of EDC Used for Correct Pathological Diagnosis

Under macroscopic examination, EDC can be both a polypoid and infiltrative lesion [15,30,31,32,33] with large areas of necrosis [15,30,32]. Occasionally, the lesion involves the lower uterine segment [15,33]. Usually, under microscopic examination, the neoplasm characteristically shows an undifferentiated component lacking any evidence of differentiation and a concomitant glandular endometrial component of a low grade (Grade 1 or Grade 2) [10]. This undifferentiated component is characterised by large necrotic areas and presents a solid growth pattern. In this component, the neoplastic cells are discohesive and monomorphic with oval nuclei, prominent nucleoli, and coarse chromatin, while mitotic figures are frequent [10].

Due to several peculiar morphological features, the correct diagnosis of this neoplasm can be exceedingly difficult and this rare and lethal malignancy may be misdiagnosed in pathological study. 

Many authors emphasized that the undifferentiated component was often located in the deeper area of the neoplasm [15,17], while the glandular endometrial differentiated component had the tendency to be located on the surface of the neoplasm. Due to this peculiar distribution of two different components, the neoplasm may be misdiagnosed as a low-grade endometrial carcinoma in endometrial curettage specimens [15,17]. Moreover, the simultaneous presence of glandular and solid areas can lead to misdiagnoses of FIGO Grade 2 or 3 endometrial adenocarcinoma [10,15,16,17]. As emphasized by some authors, for an adequate diagnosis, we should keep in mind that the solid component of Grade 3 and Grade 2 endometrial carcinoma shows a cohesive pattern, whereas the solid areas of the undifferentiated component of EDC characteristically presents cellular discohesion [10,17].

In addition, immunohistochemical studies have demonstrated that the undifferentiated component of EDC and the solid areas of Grade 3 and Grade 2 endometrial adenocarcinoma show different PAX 8 (Figure 1A), E-cadherin, cytokeratin (Figure 1B), EMA (Figure 1C), ER (Figure 1D), and PR immunoreactivity. In fact, loss of PAX 8, E-cadherin ER and PGR, focal expression of cytokeratin, and EMA can support a diagnosis of undifferentiated/dedifferentiated carcinoma over Grade 3 or Grade 2 endometrial carcinoma [22,34]. 

The dedifferentiated rhabdoid variant of EDC is characterised by the presence of an undifferentiated component which shows rhabdoid cells embedded in myxoid stroma (Figure 2A). The rhabdoid cells have eccentrically located nuclei, prominent nucleoli, and abundant and eosinophilic cytoplasm (Figure 2B). 

This variant is often misdiagnosed as a mixed Mullerian tumour (MMMT). The correct diagnosis for this subtype of EDC can be made using a specific immunohistochemical analysis which reveals SMARCA4-deficient expression in the rhabdoid component [34,35]. It is important for both the rhabdoid and the undifferentiated component of EDC to observe that these can express such neuroendocrine markers as chromogranin, synaptophysin, and neural cell adhesion molecule (CD56). In this instance, the neoplasm can mimic a neuroendocrine carcinoma and, as demonstrated by Zhou et al., may improve the prognosis of affected patients [15,32,33]. In cases with a scarce or absent well-differentiated endometrial carcinoma, the EDC undifferentiated component may be misdiagnosed as an undifferentiated endometrial sarcoma, a malignant lymphoma, or a plasmacytoma [15,22,36,37]. For a correct diagnosis in these cases, it is important to evaluate both the morphological and the immunohistochemical features. Consequently, the undifferentiated endometrial sarcoma presents more pleomorphic cells and focally spindled cells [38,39]. In addition, this neoplasm is characterised via chromosomal translocations, most commonly t(7;17)(p15;q21) involving zinc finger genes, JAZF1 and SUZ1, detectable via fluorescence in in situ hybridization (FISH) [40]. Instead, immunohistochemical analysis using Cyclin D1 is not useful for differentiating this subset of high-grade endometrial stromal sarcoma from undifferentiated endometrial carcinoma since this marker is over-expressed in both neoplasms; a molecular study is sometimes essential [41]. 

A primary lymphoma of the endometrium is exceedingly rare, and the most common histological type is diffuse large B cell lymphoma (DLBCL) [42,43]. Immunohistochemistry in this malignancy can confirm the B-cell lineage of the neoplastic cells with positive staining for other specific markers such as CD20 and CD79a [44,45], that are absent in the undifferentiated component of an EDC. Moreover, extramedullary plasmacytomas in the female genital tract are quite rare, either as solitary plasmacytomas, or as part of a disseminated multiple myeloma (MM) [36]. This disease, when it involves the uterus, can cause abnormal uterine bleeding similar to that of primary uterine carcinoma. Immunohistochemical analysis is important for diagnoses revealing immunoreactivity for CD38 and the kappa or lambda light chain [37].

Unfortunately, when uterine primary EDC shows only glandular structures and the undifferentiated component is found only in the metastases which are observed simultaneously or subsequently, a diagnosis of EDC cannot be made [10,15,21,22].

### 1.3. Genetic Alterations of EDC: Comparison with Other Subtypes of Endometrial Carcinoma and Impact on Prognosis and Treatment 

Important risk factors for survival and recurrence in endometrial cancer, such as grade, depth of invasion, presence, or absence of lymphovascular space invasion, tumour size, and lower uterine segment involvement [46,47,48,49] are particularly important and make it possible to stratify affected patients into low, intermediate, high intermediate, and high risk while they also have impact follow-up and treatment [48]. Moreover, patients at low risk, with a low grade, and low development stage are recommended for observation alone. Instead, women at a high intermediate risk and high risk (advanced stage, serous or clear cell histology, Grade 3 and deep invasion) are advised to undergo more aggressive adjuvant treatments [48]. In patients with a high risk, chemotherapy and radiotherapy, as adjuvant treatment, were suggested in several studies with recent reports from the Gynaecologic Oncology Group study 258 (GOG 258) and Post Operative Radiation Therapy (PORTEC-3) trials [50,51]. The most common cytotoxic drugs used for advanced and recurrent endometrial cancer are carboplatin and paclitaxel [52].

More recently, studies of The Cancer Genome Atlas (TCGA) Research Network which performed a genome-wide analysis of endometrial (uterine) carcinomas represented a proposal to make a reclassification of endometrial carcinoma that can distinguish distinct endometrial carcinoma types which had clinical relevance for post-surgical treatment, especially in women affected by aggressive subtypes of these neoplasms [53]. The results of this molecular analysis made it possible to identify four distinct types, namely: DNA Polymerase Epsilon (POLE) ultramutated, microsatellite instability (MSI) hypermutated, copy-number low, and copy-number high [53]. Given that whole genome sequencing to evaluate genetic alterations used by the Cancer Genome Atlas was not clinically or economically practicable on a large population base scale, many studies have suggested that more feasible techniques could be used such as immunohistochemistry [54] and Sanger or next-generation sequencing analysis [55,56,57,58]. To establish the treatments and improve the prognosis of a patient affected by endometrial carcinoma post-surgical treatment, pathologists, using immunohistochemistry and the Sanger sequencing technique, should follow a protocol study useful for classifying every neoplasm according to its molecular features. Such a classification is established on neoplastic tissue via immunohistochemistry for mismatch repair proteins, p53 expression, and the POLE sequencing technique. The absence of one or more mismatch repair of proteins’ immunohistochemical expression represents an accurate surrogate of microsatellite instability. In addition, an aberrant expression of p53 suggests the presence of Tp53 mutations. Thus, this analysis makes it possible to assign patients to four prognostic groups that can be considered surrogates of TCGA groups, such as ultramutated DNA polymerase-ε (POLE) with the best prognosis, MSI or mismatch repair deficient (MMRd) hypermutated neoplasms with an intermediate prognosis, p53 abnormal tumours with the worst prognosis, and tumours with copy-number low alterations with a good prognosis [56,57,58]. Neoplasms without defects in MMR genes, and with expression of mismatch repair proteins are named mismatched repair-proficient (MMRp). Many studies have proved that endometrial carcinoma with MMRd presents a positive correlation with the programmed cell death receptor ligand 1 protein (PD-L1) positivity [59,60], with an important impact on immunotherapeutic treatment. Programmed death receptor (PD-1) and its ligand PD-L1 are co-inhibitory trans-membrane receptors expressed on T cells that can physiologically inhibit proliferation, survival, and cytokine production of T cells, causing an immune escape [61]. This has an important impact on therapy, in fact, there are studies that have investigated PD-1 blockage for treatment in patients with advanced mismatch repair-deficient cancers, using drugs such as pembrolizumab and nivolumab both in other malignant neoplasms and endometrial carcinoma [62,63,64]. Regarding the p53 mutated group (p53mt) which reflects abnormal immunoreactivity to p53, this represents a high-risk group that has a worse prognosis with a risk of recurrences and is associated with advanced stage, Grade 3, older patient age, as well as non-endometrioid histotype, serous histotype or a ‘serous-like group’ [55]. For this group, chemotherapy and radiotherapy are important and can improve progression-free survival and overall survival rates (Figure 3) [65]. 

POLE mutated/ultramutated neoplasms on molecular analysis are characterised by the presence of POLE gene mutations which physiologically encode the catalytic subunit of DNA polymerase-ε, which, together with polymerase-δ, has an important role in DNA replication and repair alterations of DNA in eukaryotes cells [66]. The mutations in the exonuclease domain of polymerase-ε may compromise the 3′-to-5′ proofreading function, leading to the loss of replication accuracy, development of genomic instability, and consequently an ultramutated phenotype [67,68].

The POLE mutated/ultramutated group of endometrial cancers has a high tumour mutation burden, tumour neoantigen production, and tumour-infiltrating T cells. This subtype of neoplasm shows an excellent prognosis independent of other clinicopathological variables such as high-grade tumours [69,70] and the development stage [71]. The presence or absence of POLE mutation allows for a reduction in both over-treatment and under-treatment and to better delineate the prognosis of endometrial cancer [69,70,71,72]. The copy number (CN)-low (endometrioid) group is an intermediate group without a POLE mutation, abnormal expression of Tp53 and dMMR, but increased progesterone receptor expression. This suggests a basis for hormone responsiveness or in cases with CTNNB1 mutations, brachytherapy, observation, or treatment with bevacizumab [64].

To improve risk assessment for each endometrial carcinoma, it is important to integrate molecular and clinicopathological data using such feasible techniques as immunohistochemistry and Sanger or next-generation sequencing analysis [55,56,57,58,73].

The combination of pathological classification and the surrogate markers suggested by TCGA demonstrates that immunohistochemistry plays an important role in the molecular classification of endometrial carcinomas both to establish a prognosis and therapeutic strategies. Thus, it is especially important that this ancillary technique is performed accurately without any problems due to bad fixation and bad antigen preservation [74]; it is also important to consider any intra-tumoral heterogeneity that the neoplasm might present.

### 1.4. Molecular Alterations of EDC and Impact on Prognosis and Therapy

In the Cancer Genome Atlas, EDC was reported as a heterogeneous neoplasm that had a higher number of mutations, so that it could be considered a high-risk endometrial carcinoma [30,58,75,76,77,78]. Because of its rarity, poor prognosis, and association with an advanced stage at diagnosis as well as presence of higher rates of gene mutations than other high-risk endometrial cancers, there is no established treatment. In addition, due to its morphological and genetic heterogeneity, to delineate a prognosis and treatment, it is important to evaluate for each case the genomic profiling [75,76,77,78]. Many authors have reported associations between the prognosis and certain gene mutations, such as mismatch repair (MMR) gene mutations, POLE, SWI/SNF complex [77,78,79,80], and Tp53 mutations [75,81].

## 2. Mismatch Repair (MMR) Gene Mutations

EDC is generally associated with MMR deficiency and is more frequent than endometrial carcinoma [15,28,82,83,84]. Although the number of cases of EDC that have been evaluated by some authors were small [15,28,82,83,84], the data observed demonstrate that they have impacted the prognosis and therapy in this rare malignancy. In fact, it has been demonstrated that MSI tumours are more immunogenic, and that immune checkpoint inhibitors (anti PD-1/PD-L1 antibodies) are effective for some tumours, such as colorectal carcinoma, melanoma, renal, and lung carcinomas [85,86].

In addition, regarding EDC, Yokomizo et al. first demonstrated that the loss of MMR protein was observed only in the undifferentiated component [28]. In addition, Ono et al. observed that in their cases, there was MMR protein deficiency and that this was significantly associated with PD-L1 expression and the presence of tumour-infiltrating lymphocytes (CD8+), demonstrating also that EDC could be a target for immune checkpoint inhibitors. Due to the absence of PD-L1 expression in the well-differentiated component, Ono et al. suggested that for this neoplasm, the use of immune checkpoint inhibitors in combination with other conventional chemotherapeutic agents, such as paclitaxel plus carboplatin and cisplatin which could provide more satisfactory results, had an action on the growth of the well-differentiated component [84]. Immune checkpoint inhibitors, such as Pembrolizumab plus Lenvatinib have been used in KEYNOTE-146, a trial which included one case of EDC [87] or Dostarlimab, another anti PD-1 antibody, which showed an objective response in patients with recurrent or advanced MMRd Ecs [88]. Although a therapeutic strategy with a target for immune checkpoint inhibitors could improve the prognosis of patients affected by EDC, given that it is effective on the more aggressive undifferentiated component [26], it is important to keep in mind that immune checkpoint inhibitors can cause many adverse events involving multiple tissues and organs causing anaemia, lymphopenia, cutaneous rash, diarrhoea, hypoalbuminemia, dizziness, insomnia, headache, and dyspnoea [85].

## 3. POLE Domain Mutations

In the TCGA classification, the POLE mutated/ultramutated group of endometrial cancers was reported as a subtype of neoplasm that showed an excellent prognosis independently of other clinicopathological variables, such as high-grade tumours [69,70] and an advanced development stage [71]. Espinosa et al. were the first to report the presence of POLE exonuclease mutations in one case of EDC [89] and then other additional cases were reported by Rosa-Rosa et al. [76] and later again by Espinosa et al. [90]. In the series of 18 cases of EDC, Rosa-Rosa et al. observed two POLE-mutated cases. In the series of POLE-mutated undifferentiated and dedifferentiated endometrial carcinomas reported by Espinosa et al. [90], the neoplasms were more frequently stage I and the patients had a better prognosis than for other carcinomas lacking this mutational status. Of note, Espinosa et al. [90], in evaluating the survival rate, observed that patients affected by EDCs with POLE domain mutations had a better prognosis than patients affected by advanced colorectal cancer with POLE domain mutations who had a statistically significant increase in mortality despite adjuvant therapy treatment [90]. Although in the series of high-grade endometrial carcinoma with POLE mutation of Concin et al. and Yu et al., there were only a few cases of EDC and EUC; these authors suggested that in cases with concomitant POLEMut and P53abn, the patients could be managed similarly to patients with POLEmut neoplasms, since the prognosis remained good [91,92].

## 4. Switch/Sucrose Non-Fermentable (SWI/SNF) Complex

SWI/SNF complex is a family of ATP chromatin remodelling complexes present in eukaryotes. It is basically a group of proteins capable of remodelling the way DNA is packaged. In fact, this complex can stimulate ATPase activity that can destabilize histone-DNA interactions causing structural change and nucleosome rearrangement. As a result of this nucleosome rearrangement, some genes may be activated or repressed [93]. It has been demonstrated that neoplastic cells with SWI/SNF subunit mutations have disrupted chromatin structures and failed to express many genes. The subunits that are frequently mutations in mammalian malignancies are ARID1A [94], PBRM1 [95], SMARCB1 [96], SMARCA4 [97], and ARID2 [98]. In UEC/EDC, the protein products of core components of the SWI/SNF chromatin-remodelling complex that can be lost are: SMARCB1 (INI1) SMARCA2 (BRM), SMARCA4 (BRG-1), ARID1A and ARID1B for example. More commonly, in the undifferentiated component of EDC, SMARCA4 (BRG-1), ARID1A and ARID1B are inactivated, with an absence of their expression on immunohistochemical analysis [17,35,99,100,101,102].

Neoplasms with core SWI/SNF-deficiency characterise an extremely aggressive group of undifferentiated cancers which have a rapid disease progression that is refractory to conventional platinum/taxane-based chemotherapy [100]. In fact, Tessier-Cloutier et al., observed that at initial presentation, 55% of EDC with SWI/SNF deficiency had extrauterine spread in contrast to 38% of EDC with SWI/SNF-intact tumours. However, these authors observed that for prognosis, it is also important to evaluate the POLE status, given that all patients with mutated POLE showed a better prognosis with longer survival rates [100]. Until now, there are no therapeutic applications in Ec with SWI/SNF mutations. However, due to the fact that the SWI/SNF subunit mutations are observed in a wide range of malignant neoplasms [103,104,105] and that there are drugs that target these genetic alterations, the same targeted therapies could possibly enhance anti-cancer treatment effectiveness and provide new insights for therapeutic strategies in EDC. AU-15330 is a proteolysis-targeting chimera (PROTAC) degrader of the SWI/SNF ATPase subunits, SMARCA2 and SMARCA4. AU-15330 induces potent inhibition of tumour growth in xenograft models of prostate cancer and increased effectiveness of the androgen receptor enzalutamide, causing disease remission in castration-resistant prostate cancer (CRPC) models without toxicity [104].

Another therapeutic strategy that can be considered for treatment of EDC/UDC is Aurora A, which has been reported as a therapeutic target in ARID1A-deficient colorectal cancer cells [106].

In addition, it seems worth investigating the use of Tazemetostat for the treatment of EDC/UEC, since this drug is well tolerated in patients with advanced epithelioid sarcoma with a loss of INI1/SMARCCB1 [107] and its effectiveness increases in association with doxorubicin (ClinicalTrials.gov identifier: NCT042049441).

## 5. Tp53 Mutations

Tp53 mutations represent other types that can be observed in EDC [75,76,81]. In their study, Rosa-Rosa et al. observed via immunohistochemical analysis, that p53 staining was positive (aberrant) in the undifferentiated component and negative in the differentiated component, suggesting that this part of the neoplasm was developing through a ‘serous-like’ pathway [76]. Usually, the mutation of Tp53 can be evaluated using immunohistochemical analysis, which can be considered a surrogate method for molecular study. In fact, the majority of Tp53 mutations that can be observed via immunohistochemistry are missense mutations and they can be demonstrated as a detector of an overexpressed protein. However, it is important to keep in mind that there are studies which have revealed that the nonsense TP53 mutations result in an absence of immunoreactivity due to the lack of gene product [108,109]. Moreover, it is important for immunohistochemical analyses to consider that this must be performed accurately without problems due to bad fixation and bad antigen preservation [74]. In addition, it is important to consider any morphological and genetic intra-tumoral heterogeneity that the neoplasm might have [30,58,75,76,77,78]. Since Tp53 mutation represents another genetic alteration of EDC [75,76,81] agents, targeting the mutant p53 pathway could be considered for treatment of this malignancy (Table 1). Among these targeting agents, it is worth considering APR-246 (Eprenetapop). There are authors who have demonstrated that APR- 246 was well tolerated with a clinical response and remissions because it was capable of restoring wild-type p53 function in malignant cells administered in combination with azacytidine in patients with TP53-mutant myeloproliferative neoplasms, such as acute myeloid leukaemia (ClinicalTrials.gov identifier: NCT03072043) [110]. In addition, Adavosertib, which is a potent antitumor kinase inhibitor, in combination with carboplatin in advanced TP53 mutated ovarian cancer has been used at phase II (ClinicalTrials.gov identifier: NCT01164995) [111].

## 6. Conclusions

In conclusion, through our review of the literature, it is clear that EDC is an extremely aggressive neoplasm with poor prognosis. With pathological analysis, this rare malignancy can be misdiagnosed and its incidence can therefore be underestimated. However, a large number of studies in the last few years have contributed to improving pathological diagnoses through using immunohistochemical analyses. The use of immunohistochemical analysis with specific markers, which can be considered surrogates of molecular techniques, have also contributed to improving knowledge of the molecular and prognostic features of this malignancy. In addition, since EDC has a specific mutation, a therapeutic approach with potent targeting drugs could be investigated for its treatment, especially for more aggressive tumours with a worse prognosis such as those with SWI/SNF complex mutations. In our opinion, we suggest combining pathological classification and surrogate TCGA molecular classification to improve the assessment of prognosis. In addition, further molecular analyses should be investigated to establish whether genetic alterations could have an impact on immunohistochemical expression of their surrogates. With pathological examination using immunohistochemistry which can replace molecular studies, it is important that it is performed accurately without any problems due to bad fixation and bad antigen preservation [72] to avoid false negative results.

## Figures and Tables

**Figure 1 cancers-15-05155-f001:**
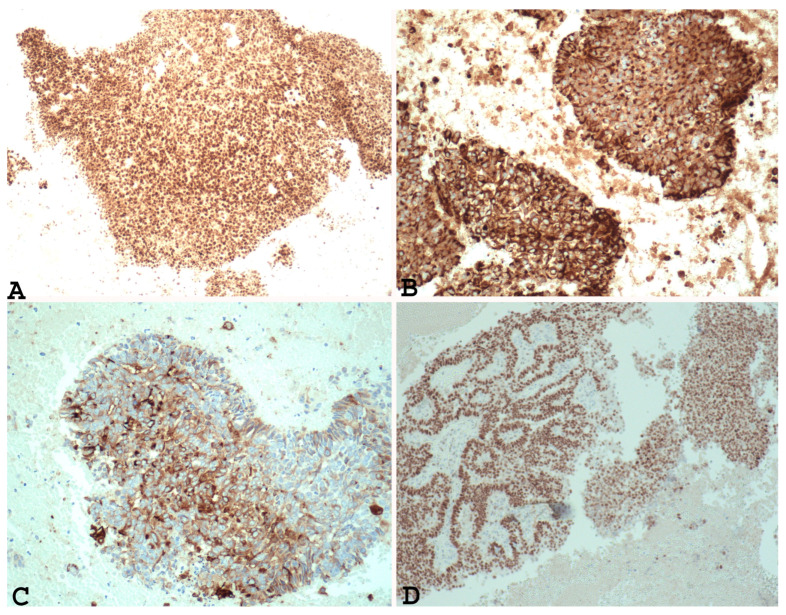
Example of Grade 3 endometrial carcinoma in endometrial curettage specimens, showing in solid component diffuse nuclear positivity to PAX-8 ((**A**) ×100), positivity to cytokeratin ((**B**) ×200), positivity to EMA ((**C**) ×200) and ER ((**D**) ×100). Note the same positivity in the glandular component and the solid component).

**Figure 2 cancers-15-05155-f002:**
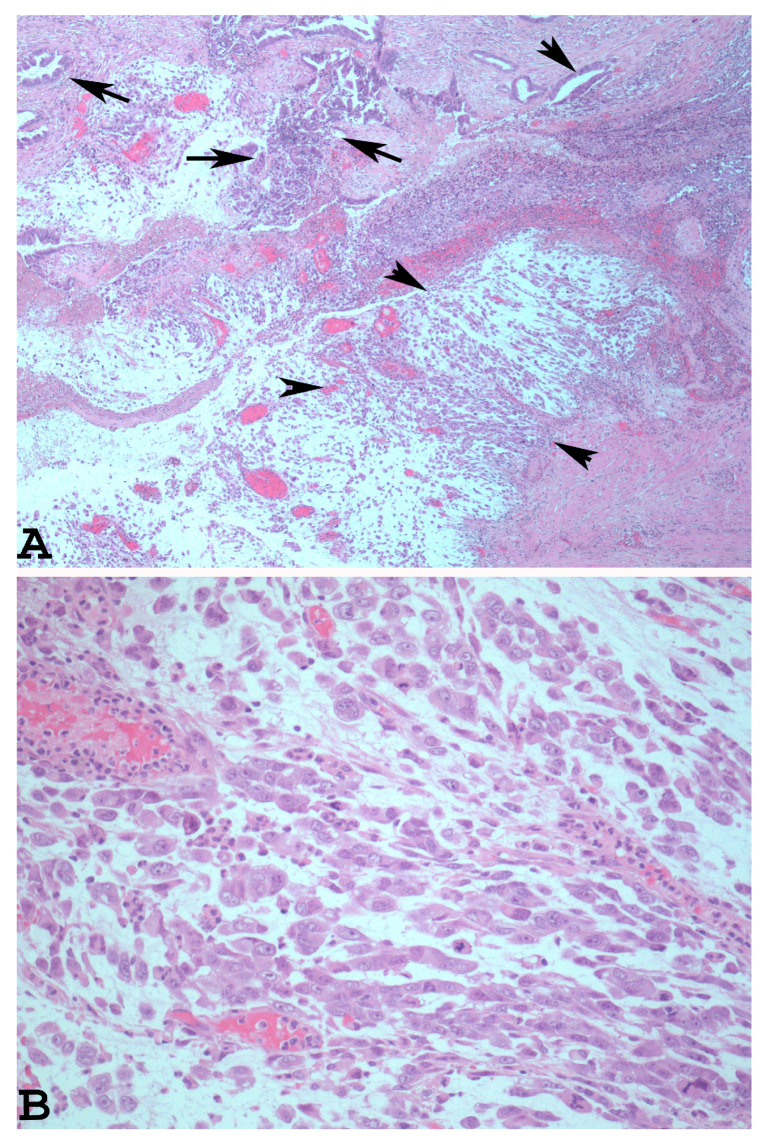
Dedifferentiated rhabdoid variant of EDC with rhabdoid cells and glandular component ((**A**): haematoxylin-eosin ×40, arrowheads: rhabdoid component. Arrows: glandular component). At higher magnification, note the eccentrically located nuclei, prominent nucleoli, and abundant and eosinophilic cytoplasm of the rhabdoid cells ((**B**): haematoxylin-eosin ×200).

**Figure 3 cancers-15-05155-f003:**
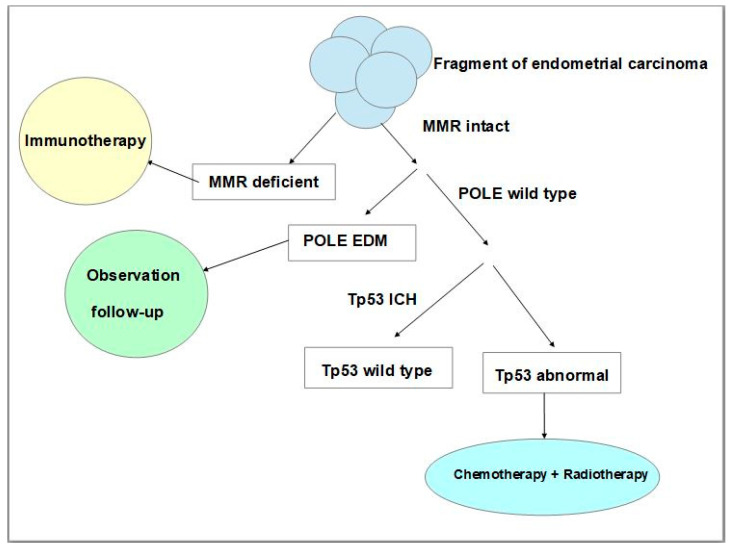
Schematic and simplified representation of the steps of molecular classification of endometrial carcinoma using immunohistochemical and Sanger sequencing techniques and the impact of therapeutic strategies. (MMR: mismatch repair, POLE: polymerase ε, IHC: immunohistochemistry, EDM: exonuclease domain mutations, Tp53: tumour protein p53).

**Table 1 cancers-15-05155-t001:** Genetic alterations in EDC and its impact on target therapy.

Genetic Alterations	Treatment for EDC or Other Neoplasms with Same Mutations
**MMR mutations** [15,28,76,82,83,84]	Immune-checkpoint inhibitors (anti PD-1/antibodies) [28]. AntiPD-L1antibodies with other chemotherapeutic agents [84]. Pembrolizumab plus Lenvatinib or Dostarlimab (anti PD-1 antibody) in recurrent or advanced MMrd Ecs [KEYNOTE-146] [87,88].
**POLE domain mutations** [76,89,90]	In EDC and EUC with concomitant POLEMut p53abn the same treatment of patients with POLEmut neoplas, since the prognosis remained good [91,92]
**SWI/SNF complex mutations** [17,35,99,100,101,102]	AU-15330 in xenograft models of prostate cancer [104]. Aurora A (ARIDIA-deficit colorectl cncer cells) [106]. Tazemetostat epithelioid sarcoma with loss of INII/SMARCCB1, with doxorubicin (Clinical Trials.gov. identifier: NCT04204944) [107].
**Tp53 mutations** [75,76,82]	APR-246 (Eprenetapop) in Tp53-mutant myeloproliferative neoplasms such as acute myeloid leukaemis (Clinical Trials.gov.identifier NCT03072043) [75,76,81,110]. APR-246 (Eprenetapop) with carboplatin in advanced Tp53 mutated ovarian cancer at phase II (Clinical Trials.gov. identifier: NCT01164995) [111].

Table Legend: Ecs: Endometrial carcinomas. EDC: Endometrial Dedifferentiated Carcinoma, MMR: Mismatch Repair, PD-1/PD-L: Programmed Death ligand 1/Programmed Death receptor, POLE: Polymerase Exonuclease-ε, SWI/SNF: Switch/Sucrose Non-Fermentable, Tp53: Tumoral p53.

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
