# Peer review of "Dedifferentiated Endometrial Carcinoma: A Rare Aggressive Neoplasm-Clinical, Morphological and Immunohistochemical Features"

_cancers, 2023, doi:10.3390/cancers15215155_

Round 1

Reviewer 1 Report

Comments and Suggestions for Authors

This is a fairly comprehensive and well presented review of a rare entity.

General language and terminology revision is due, for examples in areas there is mention of oval cytoplasm, where it should be oval nuclei.  The use of the words endometrial and endometrioid should not be used interchangeably, and wherever either is used, there needs to be due diligence in using the correct terminology.

Comments on the Quality of English Language

General revision due.

Author Response

Author's Reply to the Review Report (Reviewer 1)

Thank you so much for your for positive comment and for your useful suggestions to improve our paper

A revision of General language and terminology were made; so oval cytoplasm was replaced with oval nuclei (Page 6, Line: 132)

In many places the word endometriod was replaced with more appropriate endometrial.

Other typing errors were corrected. All changes were marked in the text.

Reviewer 2 Report

Comments and Suggestions for Authors

1. A link to Figure 1 is given in paragraph 3, but the figure itself is located at the end of the manuscript, which is inconvenient for the reader. I would recommend that the authors also provide IHC for Grade 3 and Grade 2 endometrioid adenocarcinoma for greater clarity of the differences.

2. In paragraph 4, it is better to compare genetic alterations in the form of a table.

3. I also recommend that the authors tabulate data from all clinical trials described in the review.

The study addresses the problem of coexistence of low-grade endometrial cancer and solid undifferentiated cancer, which can cause diagnostic difficulties. An accurate diagnosis of dedifferentiated endometrial carcinoma is imperative because of its poorer prognosis compared with grade 3 endometrial carcinoma, with a solid undifferentiated component that may account for up to 20% of the entire tumor.   In my opinion the research is original. This combination of cancers can often be mistaken for the more common grade 2 or 3 endometrial carcinoma; in general, this subtype of uterine cancer is often underdiagnosed.   The review provides clinical, immunohistochemical and molecular data useful for making an accurate histological diagnosis.   The authors discuss whether there are any data that could influence the prognosis or therapeutic implications of this rare and aggressive uterine neoplasm.   The conclusions are completely consistent with the evidence and arguments presented and they address the main question posed.   The links correspond to the topic of the article and are quite new.  

Author Response

Author's Reply to the Review Report (Reviewer 2)

Thank you very much for your positive comment and for useful suggestions to improve our paper.

  1. A link to Figure 1 is given in paragraph 3, but the figure itself is located at the end of the manuscript, which is inconvenient for the reader. I would recommend that the authors also provide IHC for Grade 3 and Grade 2 endometrioid adenocarcinoma for greater clarity of the differences.

We placed Figure 1 that now corresponds to Figure 2 R1 in paragraph 3. (PAGE:8)

As well as all figures were placed in the text in the paragraphs appropriate for greater clarity of the text.

We added new Figure 1R1 showing main immunohistochemical features of solid component of Grade 2 and 3 of endometrial carcinoma . (PAGE: 7) useful to differentiate solid component of dedifferentiated endometrial carcinoma and FIG 3 R1 (PAGE 12)

  1. In paragraph 4, it is better to compare genetic alterations in the form of a table.

In table 1 we listed all molecular alterations of Endometrial dedifferentiated carcinoma . (PAGE:

  1. I also recommend that the authors tabulate data from all clinical trials described in the review

In the same table 1 we showed therapeutic targets used in EDC or other neoplasm that presents the

same genetic mutations with references and clinical trials. (PAGE: 18)
